# The Impact of COVID-19 Vaccination on Inflammatory Skin Disorders and Other Cutaneous Diseases: A Review of the Published Literature

**DOI:** 10.3390/v15071423

**Published:** 2023-06-23

**Authors:** Fabrizio Martora, Teresa Battista, Angelo Ruggiero, Massimiliano Scalvenzi, Alessia Villani, Matteo Megna, Luca Potestio

**Affiliations:** Dermatology Unit, Department of Clinical Medicine and Surgery, University of Naples Federico II, Via Pansini 5, 80131 Naples, Italy; teresabattista12@gmail.com (T.B.); angeloruggiero1993@libero.it (A.R.); scalvenz@unina.it (M.S.); ali.vil@hotmail.it (A.V.); mat24@libero.it (M.M.); potestioluca@gmail.com (L.P.)

**Keywords:** psoriasis, atopic dermatitis, hidradenitis suppurativa, alopecia areata, lichen planus, vitiligo, pemphigus vulgaris, bullous pemphigoid

## Abstract

**Background:** Four vaccines have been authorized by the European Medicines Agency (EMA): viral vector-based vaccines (AstraZeneca; AZD1222 and Johnson & Johnson; Ad26.COV2. and 2 mRNA-based vaccines (Pfizer/BioNTech; BNT162b2 and Moderna; mRNA-1273). Adverse events (AEs) related to vaccination have been described in the literature. The main aim of the dermatological practice was to avoid the diffusion of COVID-19, allowing the continuity of care for patients. **Objective:** The aim of this review article is to investigate current literature regarding cutaneous reactions following COVID-19 vaccination, mainly inflammatory dermatological diseases. **Materials and methods:** Investigated manuscripts included metanalyses, reviews, letters to the editor, real-life studies, case series, and reports. **Results:** We selected a total of 234 articles involving more than 550 patients. We have divided the results section into various sub-sections to ensure greater understanding for readers. **Conclusions:** Clinicians should keep in mind the possibility of new onsets or the worsening of several dermatoses following vaccination in order to promptly recognize and treat these AEs. Certainly, vaccination should not be discouraged.

## 1. Introduction

The spreading of the Coronavirus disease 2019 (COVID-19) at the beginning of the 2020 was a worldwide challenge, strongly affecting overall health, the global economy, and lifestyles [1]. In this scenario, several measures were adopted to reduce the spread of the infection [2,3]. The main aim of the dermatological practice was to avoid the diffusion of COVID-19, allowing the continuity of care for patients [3]. This outcome led to a deep change of daily clinical practice. In particular, telemedicine represented a break of the daily routine, leading to the transition from face-to-face visits to teleconsultations in order to limit the access to hospital only for severe forms of diseases as well as to reduce the risk of infection [4,5,6]. Among the several strategies adopted, a vaccination campaign was the main one. However, several doubts were raised, regarding, for example, the quick development, the slower-than-hoped-for rollout and the uncertain duration of protection [7,8,9]. Moreover, concerns were also expressed about the mechanism of action of vaccines. Indeed, vaccines for COVID-19 are based on nucleic acid-based vaccination platforms, such as viral vector platforms, messenger ribonucleic acid and inactivated viruses [7,8,9]. Currently, four vaccines have been authorized by the European Medicines Agency (EMA): viral-vector based vaccines (AstraZeneca; AZD1222 and Johnson & Johnson; Ad26.COV2.) and two mRNA-based vaccines (Pfizer/BioNTech; BNT162b2 and Moderna; mRNA-1273) [10]. Furthermore, other countries approved other vaccines such as “CoronaVac” (Sinovac), “Convidecia” (CanSino Biologics), and “Sputnik V” (Gamaleya Research Institute) [10]. Nevertheless, the vaccination campaign was a success, allowing the COVID-19 pandemic period to be overcome, and was shown to be the most efficient weapon to control and prevent the COVID-19 pandemic, the progression of the disease, hospitalization, and mortality [11]. Currently, according to the WHO COVID-19 dashboard accessed on 11 May 2023, more than 676 million cases of COVID-19 have been reported [12,13]. Similar to other vaccines, adverse events (AEs) related to vaccination have been described, including headache, diarrhea, muscle aches, fatigue, pain or redness at the injection site, fever, chills, etc. [14]. Globally, most of these AEs were limited, self-resolving and with a duration of a few days [14]. Several cutaneous reactions have been reported following vaccination [15,16,17,18,19]. Of interest, these forms of AEs have been rarely collected in clinical trials. Indeed, global mass vaccination led to the reporting of several dermatologic reactions which were not initially recognized, getting dermatologists involved in knowing how to recognize and treat them. It is of interest that a wide spectrum of cutaneous reactions has been described [15,16,17,18,19]. However, the clinical significance of these reactions as well as the pathogenetic mechanism underlying this AE is still unknown. The aim of this review article is to investigate the current literature regarding cutaneous reactions following COVID-19 vaccination, mainly inflammatory dermatological diseases, in order to provide an overview of all the cutaneous reactions following a COVID-19 vaccine and to help clinicians to better recognize and understand these dermatological conditions.

## 2. Material and Methods

For this review article, research of the current literature was carried out in the EBSCO, PubMed, Google Scholar, Embase, MEDLINE, and Cochrane Skin databases (until 20 April 2023). The examination was performed by searching and matching the following terms: “COVID-19”, “cutaneous reactions”, “adverse events”, “vaccination”, “lichen planus”, “psoriasis”, “atopic dermatitis”, “hidradenitis suppurativa”, “bullous disorders”, “urticaria”, “atopic eczema”, “alopecia areata”, “biologics”, “biological drugs”, “rash”, “herpes”, “chilblains”, “pityriasis rosea”, “vitiligo”, “erythematous eruption” and “inflammatory skin diseases”. Manuscripts regarding other diseases such as delayed cutaneous reactions and injection site reactions were excluded. Examined manuscripts contained reviews, metanalyses, letters to the editor, case series, real-life studies, and case reports. Manuscripts were recognized, screened, and extracted for pertinent data, following the PRISMA (preferred reporting items for systematic reviews and meta-analyses) guidelines [20]. All the references were also investigated to include articles that could have been missed. Only English-language manuscripts were evaluated in our work (Figure 1).

In our review, we described dermatological reactions to COVID-19 vaccination, focusing mainly on inflammatory dermatological diseases such as psoriasis, lichen planus, atopic eczema, hidradenitis suppurativa, alopecia areata, also considering other cutaneous diseases (pityriasis rosea, herpes zoster, morphea, chilblains, pityriasis lichenoides et varioliformis acuta, Henoch–Schönlein purpura, lichen striatus and Rowell syndrome).

## 3. Results

The first research led to the discovery of 962 articles. The second research led to a skimming of 662 articles for various reasons: duplicate articles, articles not written in English. A total of 300 reports were initially found by searching the literature. Therefore, the literature review was completed by following the inclusion and exclusion criteria that we indicated in the Section 2. In the last review, in conclusion, we selected a total of 234 articles involving more than 600 patients. We have divided the Section 3 into various sub-sections to ensure greater understanding for readers; therefore, each dermatological disease will have a dedicated sub-paragraph, for example psoriasis, atopic dermatitis, hidradenitis suppurativa, lichen planus, alopecia areata, and vitiligo. Finally, we have chosen to also include a sub-paragraph with the wording “miscellaneous” where we have enclosed other dermatological pathologies found in the literature related to COVID-19 vaccination, such as morphea, chilblains, pityriasis lichenoides et varioliformis acuta, Henoch–Schönlein purpur, lichen striatus and Rowell syndrome.

### 3.1. Psoriasis

Psoriasis is a commonly occurring inflammatory skin disease that affects up to 3% of the adult population worldwide. A complex pathogenic mechanism is linked to several immune cells and cytokines, including tumor necrosis factors, interleukin (IL)-17, IL-22, and IL-23 [21,22,23,24]. Various psoriasis subtypes have been defined, including plaque, guttate, pustular, and nail psoriasis and psoriatic arthritis (PsA) [25,26,27,28]. The introduction of biologic treatments led to the development of new effective drugs [29,30,31,32,33]. Three years after the COVID-19 pandemic, the safety and efficacy of COVID-19 vaccines have been demonstrated for patients with psoriasis treated with systemic therapies [34,35,36,37]; guidelines have been drawn up that establish the vaccination times for people being treated with immunomodulatory drugs so as not to cause interference between the COVID-19 vaccine and the treatment [38]. What is increased are reports in the literature of new onsets or exacerbations of psoriasis. A distinction should be made in the case of psoriasis in that both new onsets of the condition and flare-ups of pre-existing psoriasis are reported for a total of 98 cases; 81 cases [39,40,41,42,43,44,45,46,47,48,49,50,51,52,53,54,55,56,57] involved flare-ups while the remaining 17 cases involved new onsets of psoriasis [58,59,60,61,62,63,64,65,66,67,68,69,70]. Several phenotypes have been described, such as erythrodermic, nail psoriasis, pustular psoriasis, but the plaque and guttate forms are definitely the most frequent [71,72]. Regarding the type of vaccine, most of the cases are attributed to the BNT162b2 vaccine compared to the others available; however, this data does not have much validity, as it should be emphasized that this type of vaccination was the most widely used; therefore, there is a greater risk of association of any post-vaccine dermatologic reactions. Finally, Burlando et al. reported the results of a study investigating if patients under biologics have a lower risk of psoriasis flares after COVID-19 vaccination than other psoriatic patients; their results showed that, under biologic treatment, they developed fewer psoriasis flares after COVID-19 vaccination (33.3%) than patients not under biologic treatment (66.6%) (*p* = 0.0207) [73]. Table 1 shows the characteristics of psoriatic reactions, the number of cases and the vaccines used.

### 3.2. Lichen Planus

Lichen planus (LP) is an inflammatory disease involving the skin and mucous membranes with no known cause. The lesions are pruritic, purplish papules and plaques that are mostly located on the wrists, back, and ankles [74]. To date, to the best of our knowledge, 13 cases of new onsets of the disease have been reported in the literature [75,76,77,78,79,80,81,82,83,84,85], while worsening of lichen planus after COVID-19 vaccination has been reported in three cases [84,85,86]. The underlying mechanism is certainly not known; there have been reports of new onsets of lichen planus after other vaccinations, such as that for HBV. The authors to date speculate that vaccination induces a Th1 cell response and a subsequent secretion of various cytokines that may play a key role in the development of this condition [75,76,77,78,79,80,81,82,83,84,85,86,87,88]. All articles cited with the vaccines used are given in Table 2.

### 3.3. Atopic Dermatitis/Eczema

Atopic dermatitis (AD) is a chronic inflammation that causes itchy skin with a subsequent psychosocial impact on patients and family members [89,90,91]. The most frequent clinical phenotypes in adults and adolescents are flexural eczema, head and neck eczema, and hand eczema (84.9% and 84.2%, respectively); there may also be other possible presentations, such as portrait-like dermatitis (20.1%), diffuse eczema (6.5%), nummular eczema (5.8%), prurigo nodularis (2.1%), and erythroderma (0.7%) [92,93,94]. There are very few reports in the literature linking atopic dermatitis and COVID-19 vaccination. In total there are seven reports of new onsets and 14 reports of atopic dermatitis flare or eczema [95,96,97,98,99,100,101]. Eczematous reactions, particularly those localized at the injection site, were the most misdiagnosed at the beginning of the vaccination campaign. Often, injection-site reactions began with edematous or follicular phases and then evolved into urticarial or eczematous reactions; COVID-arm became a definite entity among these reaction types [102,103,104].

There were no phenotypes described that worsened with vaccination [105,106,107,108], and there were no correlations reported between patients’ current treatments with biotech drugs such as Dupilumab that could have exacerbated atopic dermatitis. Finally, there was no correlation with a particular type of COVID-19 vaccine. All data are shown in Table 3.

### 3.4. Hidradenitis Suppurativa

Hidradenitis Suppurativa (HS) is a chronic, inflammatory, and debilitating disease of the skin.

The lesions are mainly nodules, fistulas, and/or abscesses of an inflammatory and painful nature affecting areas of the body rich in apocrine glands.

They usually occur after puberty, although there are reports in children and the elderly [109,110,111,112]. Our research found only one case of a new onset of HS associated with COVID-19 vaccination; specifically, Alexander et al. described the case of a 63-year-old patient who after the second dose of AstraZeneca COVID-19 vaccine developed abscesses in the left axilla, left side of the abdomen, and left groin [113].

The authors report that the AstraZeneca COVID-19 vaccine stimulates innate immune responses by involving multiple pattern recognition receptors, particularly Toll-like receptor 9; this could explain the correlation along with the timing of lesion appearance. Martora et al. described a case series of five patients, two men and three women, with a worsening of HS following COVID-19 vaccination: three patients with Moderna vaccine and two patients with Pfizer vaccine [114].

The important finding from this case series was that all patients regularly completed the vaccine course; the authors conclude that the mechanism is unknown, but it can be hypothesized that COVID-19 vaccine may inhibit the T helper 2 cell pathway and at the same time promote the T helper 1 cell pathway [114,115].

The researchers’ attention was directed toward the safety of the COVID-19 vaccine, particularly in patients with HS being treated with the only drug approved for the condition to date, adalimumab (anti TNF-alpha) [116,117].

Pakhchanian et al. conducted a large-scale study where they evaluated the efficacy and safety of COVID-19 vaccination in HS patients. The study involved the sampling of more than 3000 patients, and the authors conclude that patients with HS are not at any higher risk for any vaccine-related adverse outcomes [117].

Other studies were conducted with smaller samplings, where the authors’ findings described above were confirmed [118,119,120,121,122].

### 3.5. Alopecia Areata

There have been several reports of COVID-19 post-vaccination alopecia areata in recent years available in the literature [123,124,125,126,127,128,129,130,131,132,133,134,135].

There are no particular data on the type of vaccine received associated with alopecia areata; the only study in the literature on a large scale was conducted by Nguyen et al. [135]; the authors reported a total of 77 post-vaccination alopecia areata, of which 39 were new onsets and 38 were a worsening of pre-existing disease. The authors concluded their study by stating that the number was very low compared to the vaccinated population; therefore, vaccination should not be considered a risk factor in any way, and the authors urged all patients with this condition to perform the scheduled vaccination cycles [135].

### 3.6. Pemphigus Vulgaris and Bullous Pemphigoids

Rare diseases are defined by the European Community as diseases with a frequency ≤1 case per 2000 people (five cases per 10,000 people) [136]. Certainly, among these, autoimmune bullous diseases (pemphigus vulgaris and bullous pemphigoid) represent the most common. The management of these diseases during the pandemic has been very important in order not to abandon this type of patients; therefore, we can say that the role of the dermatologist in therapeutic management has been crucial [136].

In total, there are 66 reports in the literature of these two diseases. Precisely 26 concern pemphigus vulgaris [137,138,139,140,141,142,143,144,145,146,147,148,149,150,151,152], and 40 concern bullous pemphigoid [120,121,122,123,124,125,126,127,128,129,130,131,132,133,134,135]. These include reports of new onsets and reports of worsening of pre-existing disease [153,154,155,156,157,158,159,160,161,162,163,164,165,166,167,168]. All characteristics including the vaccine used are shown in Table 4.

The underlying mechanism is not known yet, but most authors write that vaccination may activate B- and T-cell immunity, triggering an autoimmune response in genetically predisposed individuals, and that this may be the most likely mechanism underlying this association. However, surely further studies are needed to confirm this hypothesis [162].

What has been crucial has been the management of these cases, where the dermatologist has been able to control these diseases without affecting the vaccine course in most cases.

### 3.7. Pityriasis Rosea

Pityriasis rosea (PR) is an exanthematous disease due to the endogenous systemic reactivation of human herpesvirus-6 (HHV-6) and/or -7 (HHV-7) [169].

Drago et al. have distinguished two different reactions: pityriasis rosea (PR) and PR-like eruptions (PR-LE) after COVID-19 vaccination [170,171]. PR-LE is not associated with human herpesvirus (HHV) 6 and/or 7 systemic reactivations, but it has a pathogenesis more similar to that of drug eruptions, for example, of captopril, barbiturates, isotretinoin [171].

Several clinical differences that have been proposed by the authors. Notably, itching is mild or absent in PR, while it is present and often intense in PR-LE; prodromal symptoms are absent in PR, while they are present in PR; herald patch is present in PR while they are in 25% of cases in PR-LE. Another differentiation concerns eosinophilia, which presents in about 42% of cases in PR-LE. The distribution of lesions involves the Christmas tree pattern on the trunk for PR, while the lesions are more confluent and distributed to the trunk, limbs and face in PR-LE [172,173]. Therapy is symptomatic for PR, while drug withdrawal is therapy for PR-LE [173].

There are 40 post COVID-19 vaccination reports in the literature [174,175,176,177,178,179,180,181,182,183,184,185,186,187,188,189,190,191]. Among them, 33 are reported after BNT162b2 vaccine, four after mRNA-1273 vaccine, three after AZD1222 vaccine, and one after Ad26.COV2 vaccine. The higher number associated with the Pfizer vaccine arises from the large number of vaccinations with this vaccine; there is no scientific correlation with this finding. Data in the literature have shown that SARS-CoV-2 infection may have played a role in the reactivation of HHV-6, -7 and EBV and, consequently, caused skin manifestations typical of pityriasis rosea [170].

The proposed hypothesis could confirm how vaccination could cause an immunosuppressive state secondary to a decrease in the amount of T lymphocytes, so a reactivation of some viruses would be explained, including pityriasis rosea [170].

Finally, a recent review showed that the association between pityriasis rosea and pityriasis rosea-like eruptions after COVID-19 vaccination is possible, but given the scarcity of studies, more studies are required to confirm this association as well as the etiology and mechanism of the disease [192].

### 3.8. Urticaria

In our review, 98 cases have been collected, also during treatment with omalizumab [95,96,101,193,194,195,196,197,198,199,200,201,202,203,204,205,206]. Of note, two types of urticaria may be distinguished: immediate and delayed, with the first as the most common, as reported by Wang et al. in their monocentric real-life study investigating cutaneous manifestations following vaccination [207] (Table 5).

### 3.9. Herpes Zoster

Varicella zoster virus (VZV) is a commonly encountered infectious disease in dermatology presenting as varicella/chickenpox or herpes zoster. Reactivation is most common in the elderly population, most commonly in individuals aged 60 years and older, the main triggers being decreased cell-mediated immunity often caused by factors such as vaccinations that lead to an aggravation of the immune system [208].

Reviewing the literature, there are 55 reports of herpes zoster post-vaccination, among them 33 post-BNT162b2 vaccine, five post-mRNA-1273 vaccine, while 19 were reported post-AZD1222 vaccine [208,209,210,211,212,213,214,215,216,217]. All authors conclude that, again, the risk of possible development of this reaction is far lower than the safety and efficacy of COVID-19 vaccination, so it is not a contraindication for the first dose or subsequent booster doses [208,209,210,211,212,213,214,215,216,217].

### 3.10. Miscellaneous

Regarding other cutaneous diseases developed following COVID-19 vaccination available or reported in literature, we have found nine cases of morphea [218,219,220,221,222,223], and 11 cases of vitiligo [224,225,226,227,228,229,230,231,232,233,234]. Other dermatoses have been described, such as chilblains, pityriasis lichenoides et varioliformis acuta, Henoch–Schönlein purpur, lichen striatus or Rowell syndrome, Sweet Syndrome, dermatomyositis, and exanthematous pustulosis. Nevertheless, these are reports of a few cases suggesting this association with the COVID-19 vaccination. Therefore, we believe that unlike the other reactions we have reported, where both the numbers and the mechanisms proposed at the base are valid, in these cases certainly further studies will be necessary to establish the true correlation [235,236,237,238,239,240,241,242,243,244].

## 4. Discussion

The COVID-19 pandemic revolutionized medical routine dermatological clinical practice [245]. Indeed, several measures were adopted to contain the spread of the infection as well as to guarantee the continuity of care, particularly for the oncodermatological field [246,247,248,249,250,251] as well as for patients undergoing biological treatments [252]. In this scenario, dermatologists were involved and forced to change their clinical routine to guarantee the continuity of care for patients with chronic inflammatory disease undergoing biological or other systemic treatments as well as to avoid the reduction in the diagnosis and treatment of several conditions, mainly melanoma and non-melanoma skin cancer [251,252,253,254]. Teledermatological services were shown to be a useful tool in this context, allowing clinicians to continuously assist patients’ diseases with promising results in terms of clinical outcomes, patient satisfaction, and treatment adherence [255,256]. Subsequently, COVID-19 vaccines were developed to overcome the pandemic period. Vaccination campaigns were successful, showing excellent results in terms of safety and efficacy [257]. Even if several concerns were raised about the effective and safe profile of vaccines, the diffusion of vaccinations allowed every doubt to be clarified [34,258]. However, several AEs were described following vaccination. Among these, several cutaneous diseases (lichen planus, psoriasis, atopic dermatitis, hidradenitis suppurativa, bullous disorders, eczema, urticaria, atopic eczema, alopecia areata, chilblains, pityriasis rosea, vitiligo, etc.) developed or exacerbated after COVID-19 vaccination have been described. Globally, most of these reactions were mild and self-limited, not requiring medical attention. For the same reason, most of these cutaneous AEs were not reported, as patients tended to self-medicate, without seeking medical advice. To date, dermatologic reports after the first vaccination are at one case in a million. At the same time, it is still unclear whether repeated doses of vaccination according to the schedule can affect the decrease in these events. In this context, we performed a review article with the aim of reporting data on cutaneous reactions described following COVID-19 vaccination in order to offer a wide perspective as well as trying to underline possible pathogenetic factors. As regards their pathogenesis, cutaneous reactions were reported following vaccination with both viral vector-based and mRNA vaccines, suggesting that the pathogenetic mechanism underlying the cutaneous reaction is not directly connected with the mechanism of action of the vaccine itself. Certainly, mRNA vaccines seem to be more commonly related to cutaneous reactions. However, mRNA vaccines have been the more used, as they were previously authorized and administered worldwide. Moreover, cutaneous reactions were reported following the first, the second and the third dose of vaccines, highlighting that each dose may be related to the development of AEs. Thus, more studies are required to recognize the pathogenetic mechanisms related to cutaneous reactions following COVID-19 vaccination in order to identify “at-risk” patients and to adopt preventive strategies. Certainly, further epidemiological studies will help to clarify if the percentage of cutaneous reactions following vaccination is significantly higher in one of the two types of vaccines, with clinical implications. Despite not being specifically investigated, local injection-site reactions were the commonest cutaneous vaccine-related AEs collected. As regards cutaneous inflammatory disease, there are several reports of their new onset or worsening following COVID-19 vaccination. However, the chronic relapsing course of these diseases does not allow the casual correlation to be excluded in most of the cases. Thus, the exact link between vaccination and cutaneous AE has not been elucidated. Of interest, viral reactivations (e.g., EBV reactivation, other herpesviruses) have also been reported in a minority of cases following COVID-19 vaccination [258,259,260,261].

To sum up, our review manuscript investigated several dermatoses developed or exacerbated following COVID-19 vaccination. However, the causality of the temporal association between the administration of the vaccine and the development of skin reactions cannot be ruled out. As regards the dose of vaccine, cutaneous reactions were described following both the first and the second dose of vaccine, while cutaneous AEs were also reported after the third (booster) dose of vaccination. In our opinion, clinicians should keep in mind the possibility of the development or new onset of cutaneous AEs following vaccination, regardless of the dose and the mechanism of action of the vaccine.

### Strengths and Limitations

The main strengths of this review article are the systematic method used for the literature research as well as the elevated number of investigated articles and analyzed cutaneous reactions. However, the main limitations must be discussed. First of all, only the four vaccines approved by EMA have been considered in this manuscript, excluding cutaneous reactions related to other COVID-19 vaccines. Furthermore, many manuscripts reporting registry-based studies did not allow direct correlations to be established between cutaneous reactions and the type of vaccine. Post-vaccination dermatology reports often contain case reports or small case series; there is a lack of guidelines for the management of these manifestations, as each case certainly needs to be treated according to the pathology, and treatment may depend on various factors. In addition, the causal temporal correlation between cutaneous reaction and vaccination cannot be ruled out in most of the cases.

Moreover, dermatological conditions exacerbated or developed following COVID-19 vaccines are usually mild, and patients do not seek medical attention, reducing the possibility to point out the exact incidence of these reactions.

Finally, our assumptions, mainly in the discussion, must be taken simply as suggestions and not as definite proposals, as our work has not had the support of meta-analysis, which may allow our results to be generalized.

## 5. Conclusions

Vaccination campaigns were the main weapon to overcome the COVID-19 pandemic period. With the progress of vaccination programs, several cutaneous reactions have been reported; most of these were not previously assessed in clinical trials. Fortunately, the percentage of these AEs is extremely low if compared with the number of vaccines administered. In our opinion, other cutaneous reactions related to COVID-19 vaccination will be described. Furthermore, the pathogenetic mechanisms linking vaccination and skin reactions should be investigated to identify “at-risk” patients and adopt preventative measures. To sum up, clinicians should keep in mind the possibility of new onsets or the worsening of several dermatoses following vaccination in order to promptly recognize and treat these AEs. Certainly, vaccination should not be discouraged.

## Figures and Tables

**Figure 1 viruses-15-01423-f001:**
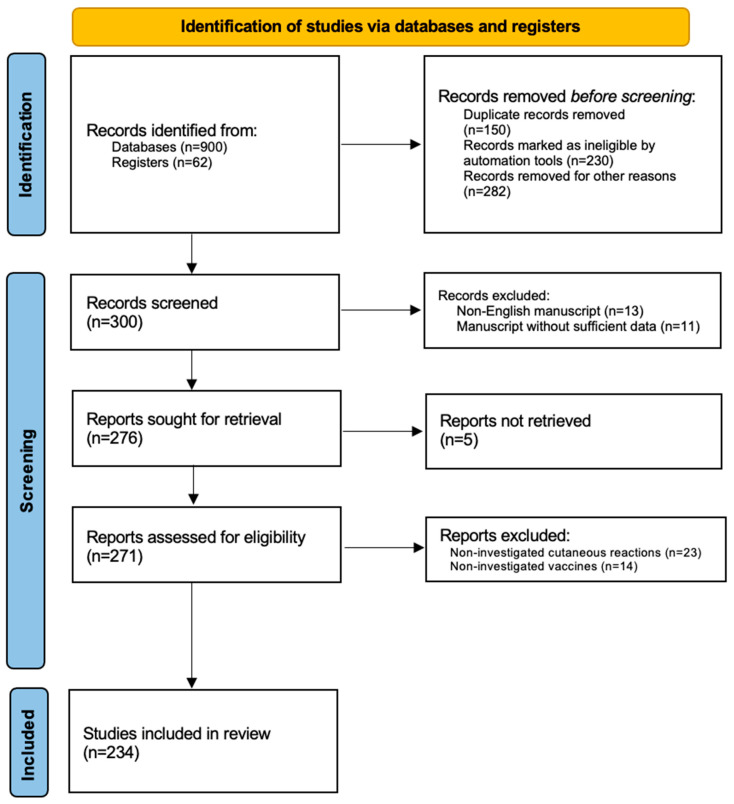
PRISMA flow-chart.

**Table 1 viruses-15-01423-t001:** Features of Psoriasis after COVID-19 vaccination.

Reaction	Number of Cases	Authors and Number of Cases	Phenotype of Psoriasis	Total Amount of Vaccines Used
Flare of Psoriasis	81	Huang et al. (15), Sotiriou et al. (14), Koumaki et al. (12), Megna et al. (11), Wei et al. (6), Ruggiero et al. (4), Durmaz et al. (2), Tran et al. (2), Piccolo et al. (2), Bostan et al. (1), Nagrani et al. (1), Pavia et al.(1), Durmus et al. (1), Fang et al. (1), Krajewski et al. (1), Trepanowski et al. (1), Mieczkowska et al. (1), Lopez et al. (1), Perna et al. (1), Tsunoda et al. (1), Nia et al. (1), Pesqué et al. (1).	Plaque Psoriasis (68%) Nail Psoriasis (2%) Erythrodermic Psoriasis (5%) Annular Psoriasis (3%) Guttate Psoriasis (22%)	BNT162b2: 43 mRNA-1273: 17 AZD1222: 21 Ad26.COV2: 0
New onset of Psoriasis	17	Tran et al. (3), Ouni et al.(2), Nagrani et al. (1), Song et al. (1), Frioui et al. (1), Cortonesi et al. (1), Lehmann et al.(1), Elamin et al. (1), Wei et al. (1), Lamberti et al.(1), Romagnuolo et al. (1), Ruggiero et al.(1), Ricardo et al. (1), Pesqué et al. (1).	Plaque Psoriasis (52%) Nail Psoriasis (5%) Erythrodermic Psoriasis (5%) Annular Psoriasis (3%) Guttate Psoriasis (35%)	BNT162b2: 10 mRNA-1273: 3 AZD1222: 3 Ad26.COV2: 1

**Legend:** BNT162b2, Pfizer mRNABNT162b2; mRNA-1273, Moderna mRNA-1273; AZD1222, AstraZeneca-Oxford AZD1222; Ad26.COV2, Johnson & Johnson Ad26.COV2. S.

**Table 2 viruses-15-01423-t002:** Features of Lichen Planus after COVID-19 vaccination.

Reaction	Number of Cases	Authors and Number of Cases	Total Amount of Vaccines Used
New Onset Lichen Planus	13	Merhy et al. (1), Kato et al. (1), Diab et al. (1), Zagaria et al. (1), Awada et al. (1), Picone et al. (1), Hlaca et al. (1), Zengarini et al.(1), Masseran et al.(1), Gamonal et al. (1), Alrawashdeh et al. (1), Shakoei et al. (1).	BNT162b2: 8 mRNA-1273: 1 AZD1222: 7 Ad26.COV2: 0
Flares of Lichen planus	3	Hiltun et al. (1), Herzum et al. (1), Hlaca et al. (1).	BNT162b2: 8 mRNA-1273: 1 AZD1222: 7 Ad26.COV2: 0

**Legend:** BNT162b2, Pfizer mRNABNT162b2; mRNA-1273, Moderna mRNA-1273; AZD1222, AstraZeneca-Oxford AZD1222; Ad26.COV2, Johnson & Johnson Ad26.COV2. S.

**Table 3 viruses-15-01423-t003:** Features of Atopic Dermatitis/Eczema following COVID-19 vaccination.

Reaction	Number of Cases	Authors and Number of Cases	Total Amount of Vaccines Used
New Onset Atopic Dermatitis/Eczema	7	Rerknimitr et al. (3), Holmes et al. (1), Leasure et al. (1), Bekkali et al. (1), Larson et al. (1).	BNT162b2: 3 mRNA-1273: 1 AZD1222: 3 Ad26.COV2: 0
Flares of Atopic Dermatitis/Eczema	14	Potestio et al. (11), Leasure et al. (1), Niebel et al. (1), Larson et al. (1).	BNT162b2: 8 mRNA-1273: 3 AZD1222: 3 Ad26.COV2: 0

**Legend:** BNT162b2, Pfizer mRNABNT162b2; mRNA-1273, Moderna mRNA-1273; AZD1222, AstraZeneca-Oxford AZD1222; Ad26.COV2, Johnson & Johnson Ad26.COV2. S.

**Table 4 viruses-15-01423-t004:** Features of Pemphigus Vulgaris and Bullous Pemphigoid after COVID-19 vaccination.

Reaction	Number of Cases	Authors and Number of Cases	Total Amount of Vaccines Used
Pemphigus Vulgaris	26	Martora et al. (7), Zou et al. (3), Gui et al. (2), Rouatbi et al. (2), Aryanian et al. (1), Koutlas et al. (1), Knechtl et al. (1), Ong et al. (1), Yıldırıcı et al. (1), Singh et al. (1), Norimatsu et al. (1), Agharbi et al. (1), Almasi-Nasrabadi et al. (1), Corrá et al. (1), Solimani et al. (1).	BNT162b2: 15 mRNA-1273: 6 AZD1222: 5 Ad26.COV2: 0
Bullous Pemphigoid	40	Maronese et al. (21), Maronese et al. (3), Hali et al. (3), Gambichler et al. (2), Shanshal et al. (1), Desai et al. (1), Fu et al. (1), Alshammari et al. (1), Hung et al. (1), Pauluzzi et al. (1), Dell’Antonia et al. (1), Pérez-López et al. (1), Agharbi et al. (1), Young et al. (1), Nakamura et al. (1).	BNT162b2: 29 mRNA-1273: 5 AZD1222: 6 Ad26.COV2: 0

**Legend:** BNT162b2, Pfizer mRNABNT162b2; mRNA-1273, Moderna mRNA-1273; AZD1222, AstraZeneca-Oxford AZD1222; Ad26.COV2, Johnson & Johnson Ad26.COV2. S.

**Table 5 viruses-15-01423-t005:** Features of urticarial rashes after COVID-19 vaccination.

Reaction	Number of Cases	Authors and Number of Cases	Total Amount of Vaccines Used
Urticarial Rashes	98	Magen et al. (39), Potestio et al. (15), Rerknimitr et al. (12), Riad et al. (10), Sidlow et al. (3), Peigottu et al. (2), Niebel et al. (2), McMahon et al. (2), Holmes et al. (2), Fernandez-Nieto et al. (2), Bianchi et al. (2), Corbeddu et al. (2), Baraldi et al. (1), Choi et al. (1), Patruno et al. (1), Burlando et al. (1), Thomas et al. (1).	BNT162b2: 15 mRNA-1273: 6 AZD1222: 5 Ad26.COV2: 0

**Legend:** BNT162b2, Pfizer mRNABNT162b2; mRNA-1273, Moderna mRNA-1273; AZD1222, AstraZeneca-Oxford AZD1222; Ad26.COV2, Johnson & Johnson Ad26.COV2. S.

## Data Availability

Data sharing not applicable to this article as no datasets were generated or analyzed during the current study.

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
