# Peer review of "The Impact of COVID-19 Vaccination on Inflammatory Skin Disorders and Other Cutaneous Diseases: A Review of the Published Literature"

_viruses, 2023, doi:10.3390/v15071423_

Round 1

Reviewer 1 Report

In their work “The impact of COVID-19 vaccination on inflammatory skin disorders: A review of the published literature” the authors provide an updated and intriguing description of dermatological reactions to Covid-19 vaccination, focusing mainly on inflammatory dermatological diseases.

Overall, the manuscript is well written and well-structured and provides a clearcut and informing review of relevant dermatological reactions to COVID-19 vaccination.

I suggest only minor revisions and strongly recommend publication.

MINOR REVISIONS:

As reported by the authors in lines 65-69 “The aim of this review article is to investigate
current literature regarding cutaneous reactions developed following Covid-19 vaccination, mainly inflammatory dermatological diseases, in order to provide an overview of all the cutaneous reactions following Covid-19 vaccine and to help clinicians to better recognize and understand these dermatological conditions.

Please keep with this aim also in the Materials and Methods section, where, in lines 88-90, the authors are more restrictive,  stating that “Psoriasis, lichen planus, atopic eczema, hidradenitis suppurativa, alopecia areata, pityriasis rosea, herpes zoster, morphea, chilblains, pityriasis lichenoides et varioliformis  acuta, Henoch-Schonlein purpur, lichen striatus and Rowell syndrome are the dermatologic disorders evaluated in our review”.

I would suggest just to say you will “describe dermatological reactions to Covid-19 vaccination, focusing mainly on  inflammatory dermatological diseases, such as psoriasis, lichen planus and so on”.

I really appreciate the effort of reassuming and commenting the most relevant dermatological issues linked to COVID 19 vaccination in just one work, which is of great interest for readers.

However, though I understand the theme is very wide, I noticed that at present, some relevant articles were not included in the study. Therefore I suggest either to include the latter or to focus more deeply on exclusion criteria in the Materials and Methods section:

 (e.g. Khan I, Elsanousi AA, Shareef AM, Tebha SS, Arif A, Gul S. Manifestation of pityriasis rosea and pityriasis rosea-like eruptions after Covid-19 vaccine: A systematic review. Immun Inflamm Dis. 2023 Apr;11(4):e804. doi: 10.1002/iid3.804;

Burlando M, Herzum A, Cozzani E, Parodi A. Psoriasis flares after COVID-19 vaccination: adherence to biologic therapy reduces psoriasis exacerbations: a case-control study. Clin Exp Vaccine Res. 2023 Jan;12(1):80-81. doi: 10.7774/cevr.2023.12.1.80. Epub 2023 Jan 31.

Genco L, Cantelli M, Noto M, Battista T, Patrì A, Fabbrocini G, Vastarella M. Alopecia Areata after COVID-19 Vaccines. Skin Appendage Disord. 2023 Mar;9(2):141-143. doi: 10.1159/000528719. Epub 2023 Jan 19.

Wang R, Mathes S, Claussen C, Biedermann T, Brockow K. Cutaneous reactions following COVID-19 vaccination assessed by dermatologists: a single-institutional study in Germany. J Dtsch Dermatol Ges. 2023 Mar;21(3):255-262. doi: 10.1111/ddg.14987. Epub 2023 Mar 9. PMID: )

Also, I suggest a section regarding miscellaneous viral reactivations, would improve the manuscript,  possibly discussing also infections reported in a minority of cases (e.g. EBV reactivation, other herpesviruses) for example as a heading for the paragraphs regarding pityriasis rosea and herpes zoster

(e.g. Navarro-Bielsa A, Gracia-Cazaña T, Aldea-Manrique B, Abadías-Granado I, Ballano A, Bernad I, Gilaberte Y. COVID-19 infection and vaccines: potential triggers of Herpesviridae reactivation. An Bras Dermatol. 2023 May-Jun;98(3):347-354. doi: 10.1016/j.abd.2022.09.004. Epub 2023 Feb 10. ;

Herzum A, Trave I, D'Agostino F, Burlando M, Cozzani E, Parodi A. Epstein-Barr virus reactivation after COVID-19 vaccination in a young immunocompetent man: a case report. Clin Exp Vaccine Res. 2022 May;11(2):222-225. doi: 10.7774/cevr.2022.11.2.222. Epub 2022 May 31. PMID: 35799871; PMCID: PMC9200649.)

Also, cutaneous delayed hypersensitivities and urticarial rashes deserve a separate paragraph, possibly also mentioning immediate cutaneous reactions, as these types of reactions were  frequently reported

 (e.g. Hanoodi M, Logee DO K. A Rare Case of Delayed Hypersensitivity Following COVID-19 Booster Necessitating Treatment With Dupilumab. Cureus. 2023 Apr 13;15(4):e37544. doi: 10.7759/cureus.37544. ;

 Burlando M, Herzum A, Cozzani E, Parodi A. Acute urticarial rash after COVID-19 vaccination containing Polysorbate 80. Clin Exp Vaccine Res. 2021 Sep;10(3):298-300. doi: 10.7774/cevr.2021.10.3.298. Epub 2021 Sep 30. ;

Wang R, Mathes S, Claussen C, Biedermann T, Brockow K. Cutaneous reactions following COVID-19 vaccination assessed by dermatologists: a single-institutional study in Germany. J Dtsch Dermatol Ges. 2023 Mar;21(3):255-262. doi: 10.1111/ddg.14987. Epub 2023 Mar 9.)

).

I suggest only minor language editing revisions:

 - l. 286 write “chilblains” instead of “chilbrains

- l. 105 write “Henoch–Schönlein purpura” instead of  “Henoch-Schonlein purpur”

Author Response

In their work “The impact of COVID-19 vaccination on inflammatory skin disorders: A review of the published literature” the authors provide an updated and intriguing description of dermatological reactions to Covid-19 vaccination, focusing mainly on inflammatory dermatological diseases.

Overall, the manuscript is well written and well-structured and provides a clearcut and informing review of relevant dermatological reactions to COVID-19 vaccination.

I suggest only minor revisions and strongly recommend publication.

Author response: thanks for appreciating our manuscript.

MINOR REVISIONS:

As reported by the authors in lines 65-69 “The aim of this review article is to investigate 
current literature regarding cutaneous reactions developed following Covid-19 vaccination,mainly inflammatory dermatological diseases, in order to provide an overview of all the cutaneous reactions following Covid-19 vaccine and to help clinicians to better recognize and understand these dermatological conditions. “Please keep with this aim also in the Materials and Methods section, where, in lines 88-90, the authors are more restrictive,  stating that “Psoriasis, lichen planus, atopic eczema, hidradenitis suppurativa, alopecia areata, pityriasis rosea, herpes zoster, morphea, chilblains, pityriasis lichenoides et varioliformis  acuta, Henoch-Schonlein purpur, lichen striatus and Rowell syndrome are the dermatologic disorders evaluated in our review”.I would suggest just to say you will “describe dermatological reactions to Covid-19 vaccination, focusing mainly on  inflammatory dermatological diseases, such as psoriasis, lichen planus and so on”.

Author response: thanks for the comment. Methods section has been rewritten as suggested. 

I really appreciate the effort of reassuming and commenting the most relevant dermatological issues linked to COVID 19 vaccination in just one work, which is of great interest for readers.

However, though I understand the theme is very wide, I noticed that at present, some relevant articles were not included in the study. Therefore I suggest either to include the latter or to focus more deeply on exclusion criteria in the Materials and Methods section:

 (e.g. Khan I, Elsanousi AA, Shareef AM, Tebha SS, Arif A, Gul S. Manifestation of pityriasis rosea and pityriasis rosea-like eruptions after Covid-19 vaccine: A systematic review. Immun Inflamm Dis. 2023 Apr;11(4):e804. doi: 10.1002/iid3.804;

Burlando M, Herzum A, Cozzani E, Parodi A. Psoriasis flares after COVID-19 vaccination: adherence to biologic therapy reduces psoriasis exacerbations: a case-control study. Clin Exp Vaccine Res. 2023 Jan;12(1):80-81. doi: 10.7774/cevr.2023.12.1.80. Epub 2023 Jan 31.

Genco L, Cantelli M, Noto M, Battista T, Patrì A, Fabbrocini G, Vastarella M. Alopecia Areata after COVID-19 Vaccines. Skin Appendage Disord. 2023 Mar;9(2):141-143. doi: 10.1159/000528719. Epub 2023 Jan 19.

Wang R, Mathes S, Claussen C, Biedermann T, Brockow K. Cutaneous reactions following COVID-19 vaccination assessed by dermatologists: a single-institutional study in Germany. J Dtsch Dermatol Ges. 2023 Mar;21(3):255-262. doi: 10.1111/ddg.14987. Epub 2023 Mar 9. PMID: )

 Author response: thanks for the suggestion. The suggested manuscripts have been added to our review as well as Methods section has been reorganized as suggested. 

Also, I suggest a section regarding miscellaneous viral reactivations, would improve the manuscript,  possibly discussing also infections reported in a minority of cases (e.g. EBV reactivation, other herpesviruses) for example as a heading for the paragraphs regarding pityriasis rosea and herpes zoster 

(e.g. Navarro-Bielsa A, Gracia-Cazaña T, Aldea-Manrique B, Abadías-Granado I, Ballano A, Bernad I, Gilaberte Y. COVID-19 infection and vaccines: potential triggers of Herpesviridae reactivation. An Bras Dermatol. 2023 May-Jun;98(3):347-354. doi: 10.1016/j.abd.2022.09.004. Epub 2023 Feb 10. ;

Herzum A, Trave I, D'Agostino F, Burlando M, Cozzani E, Parodi A. Epstein-Barr virus reactivation after COVID-19 vaccination in a young immunocompetent man: a case report. Clin Exp Vaccine Res. 2022 May;11(2):222-225. doi: 10.7774/cevr.2022.11.2.222. Epub 2022 May 31. PMID: 35799871; PMCID: PMC9200649.)

 Author response: thanks for the comment. We briefly discuss the possibility of viral reactivations in our discussion as this topic is not the aim of our manuscript.

Also, cutaneous delayed hypersensitivities and urticarial rashes deserve a separate paragraph, possibly also mentioning immediate cutaneous reactions, as these types of reactions were  frequently reported

 (e.g. Hanoodi M, Logee DO K. A Rare Case of Delayed Hypersensitivity Following COVID-19 Booster Necessitating Treatment With Dupilumab. Cureus. 2023 Apr 13;15(4):e37544. doi: 10.7759/cureus.37544. ;

 Burlando M, Herzum A, Cozzani E, Parodi A. Acute urticarial rash after COVID-19 vaccination containing Polysorbate 80. Clin Exp Vaccine Res. 2021 Sep;10(3):298-300. doi: 10.7774/cevr.2021.10.3.298. Epub 2021 Sep 30. ;

Wang R, Mathes S, Claussen C, Biedermann T, Brockow K. Cutaneous reactions following COVID-19 vaccination assessed by dermatologists: a single-institutional study in Germany. J Dtsch Dermatol Ges. 2023 Mar;21(3):255-262. doi: 10.1111/ddg.14987. Epub 2023 Mar 9.)

).

  Author response: thanks for the comment. An urticaria section has been added. However, we did not report cutaneous delayed hypersensitivities as specified in methods.

Comments on the Quality of English Language

I suggest only minor language editing revisions:

 - l. 286 write “chilblains” instead of “chilbrains”

- l. 105 write “Henoch–Schönlein purpura” instead of  “Henoch-Schonlein purpur”

 Author response: thanks for the comment. Minor errors have been corrected.

Reviewer 2 Report

Dear author,

First of all, I would like to congratulate you for your work, writing a revision is always daring and difficult.

Line 57, put etc or a dot instead of ….

Line 59 and 63 are redundant and you are repeating of in the start of two consecutive sentences. It is of interest, or it is noticeable.

Line 91: please avoid auto reference when you are writing, do not put your review, you should put in this review.

Line 92: That are indicated, you should use the passive form in formal English.

Line 97: again, the same.

Line 98: Results were divided, instead of we divided.

Line 117: What is increased, used passive.

Line 123: could you put a percentage?

Table 1: You should re-edit the table, there is a lot of authors in the column of authors, try to summarize and simplify. When you put the type of psoriasis, please add the total number out of what, for example, plaque psoriasis x/ 81 (68%).

Line 139 and 142: “To date” repeated very close.

Table 2: Too many authors with one case. Try to summarize, maybe you can fusion table 1, 2 and 3 as they have 2 files in one of them. The total amount of vaccines used does not add any information as they are all used.

Line 169: maybe concluded that there is no correlation is too strong, while a statistical analysis has not been performed. You should be less categorical and only describe.

Hidradenitis suppurativa: too much discussion in comparison to other dermatosis more affected by vaccination. The distribution of the manuscript looks disbalance.

Table 4: too many authors, try to sum up, for example using only the reference.

Line 253: change the style, from there are several clinical to several clinical have been proposed.

From my view, herpes zoster and herpes simple needs more attention, as we have seen so many reactivations in practice.

Discussion: please separate in paragraphs.

You should add a conclusion and why not, a recommendation.

Your manuscript is original and of interest, nevertheless, needs English editing style improvement and in such important dermatosis as psoriasis or atopic dermatitis you should add more discussion or propose mechanism adding something new, original and interesting.

Best

Author Response

First of all, I would like to congratulate you for your work, writing a revision is always daring and difficult.

Author Response: Thank you very much we are honored by your comment

Line 57, put etc or a dot instead of ….

Line 59 and 63 are redundant and you are repeating of in the start of two consecutive sentences. It is of interest, or it is noticeable.

Line 91: please avoid auto reference when you are writing, do not put your review, you should put in this review.

Line 92: That are indicated, you should use the passive form in formal English.

Line 97: again, the same.

Line 98: Results were divided, instead of we divided.

Line 117: What is increased, used passive.

Line 123: could you put a percentage?

Author response: Thank you for all these comments, we have edited all the errors you pointed out to us

Table 1: You should re-edit the table, there is a lot of authors in the column of authors, try to summarize and simplify. When you put the type of psoriasis, please add the total number out of what, for example, plaque psoriasis x/ 81 (68%).

Author response: Thank you for the comment, we wanted to point out how the guidelines of the paper require for this type of articles at least 4000 words with tables and figures , that's the reason for the many tables, plus we think they are essential to make a summary of what is written in the paragraphs, we tried to edit as much as possible to make it more understandable for readers. 

Line 139 and 142: “To date” repeated very close.

Author response : Ok

Table 2: Too many authors with one case. Try to summarize, maybe you can fusion table 1, 2 and 3 as they have 2 files in one of them. The total amount of vaccines used does not add any information as they are all used.

Author response: Thank you for the comment, we wanted to point out how the guidelines of the paper require for this type of articles at least 4000 words with tables and figures , that's the reason for the many tables, plus we think they are essential to make a summary of what is written in the paragraphs, we tried to edit as much as possible to make it more understandable for readers. 

Line 169: maybe concluded that there is no correlation is too strong, while a statistical analysis has not been performed. You should be less categorical and only describe.

 Author response: Ok we did it, thanks a lot

Hidradenitis suppurativa: too much discussion in comparison to other dermatosis more affected by vaccination. The distribution of the manuscript looks disbalance.

Author response: We don't think it is unbalanced in fact there are few cases reported so it can be a very interesting topic for the future

Table 4: too many authors, try to sum up, for example using only the reference.

Author response: Thank you for the comment, we wanted to point out how the guidelines of the paper require for this type of articles at least 4000 words with tables and figures , that's the reason for the many tables, plus we think they are essential to make a summary of what is written in the paragraphs, we tried to edit as much as possible to make it more understandable for readers. 

Line 253: change the style, from there are several clinical to several clinical have been proposed.

Author response: Ok we did it

From my view, herpes zoster and herpes simple needs more attention, as we have seen so many reactivations in practice.

Author response: We have added other case or case series, thanks for suggestions

Discussion: please separate in paragraphs.

You should add a conclusion and why not, a recommendation.

Author response: This is already in the manuscript, there are 3 final paragraphs, discussion limitations and conclusions

Your manuscript is original and of interest, nevertheless, needs English editing style improvement and in such important dermatosis as psoriasis or atopic dermatitis you should add more discussion or propose mechanism adding something new, original and interesting.

Author response: We added various news items , unfortunately to date the mechanisms are still poorly known so we reported everything in the literature arguing the proposed mechanisms. We have revised all the English language in the text

Reviewer 3 Report

In this manuscript for a review article, authors summarize current evidence on skin reactions to Covid-19 vaccines. The topic of the manuscript is important, however, there are several issues to be addressed by the authors:

Major issues:

1.       The manuscript is too lengthy in general, that renders it difficult for the authors to comprehend the text, it should be more concise

2.       Prisma diagram should be included

3.       It is not clear which disoders do the author classify as inflammatory, if morphea is inflammatory other autoimmune disorders should be as well

4.       The work is incomplete, as papers relevant to the topic have not been cited, such as :

Majid I, Mearaj S. Sweet syndrome after Oxford-AstraZeneca COVID-19 vaccine (AZD1222) in an elderly female. Dermatol Ther. 2021 Nov;34(6):e15146. doi: 10.1111/dth.15146. Epub 2021 Oct 7. PMID: 34590397; PMCID: PMC8646808.

Agaronov A, Makdesi C, Hall CS. Acute generalized exanthematous pustulosis induced by Moderna COVID-19 messenger RNA vaccine. JAAD Case Rep. 2021 Oct;16:96-97. doi: 10.1016/j.jdcr.2021.08.013. Epub 2021 Aug 27. PMID: 34466640; PMCID: PMC8393513.

Paolino G, Di Nicola MR, Cantisani C, Mercuri SR. Pityriasis rosea infection in a COVID-19 patient successfully treated with systemic steroid and antihistamine via telemedicine: Literature update of a possible prodromal symptom of an underlying SARS-CoV-2 infection. Dermatol Ther. 2021 Jul;34(4):e14972. doi: 10.1111/dth.14972. Epub 2021 May 24. PMID: 33993616; PMCID: PMC8209955.

Cantisani C et al. Cutaneous Reactions to COVID-19 Vaccines in a Monocentric Study: A Case Series. J Clin Med. 2022 Jun 30;11(13):3811. doi: 10.3390/jcm11133811. PMID: 35807096; PMCID: PMC9267144.

Camargo Coronel et al. Dermatomyositis post vaccine against SARS-COV2. BMC Rheumatol. 2022 Apr 1;6(1):20. doi: 10.1186/s41927-022-00250-6. PMID: 35361289; PMCID: PMC8970647.

5.       The selection of the searched keywords seem a bit random

6.       In is not questioned if a vaccine reaction is reported somewhere it could be not related to the vaccine, as the majority of the population received vaccines, causality is sometimes hard to prove

7. There should be more comprehensive tables and figures, as overall it is difficult to draw the main findings of the paper

Minor issues:

1.       Keywords do not include Covid-19

2.       Numbers should be sound, such as in the abstract, why skin reactions of about 550 patients were included, is not there an exact number?

There are several typos in the text and the use of spaces are not accurate, please see even the abstract

Author Response

In this manuscript for a review article, authors summarize current evidence on skin reactions to Covid-19 vaccines. The topic of the manuscript is important, however, there are several issues to be addressed by the authors:

Major issues:

  1. The manuscript is too lengthy in general, that renders it difficult for the authors to comprehend the text, it should be more concise

Author response: thanks for appreciating our manuscript. The text has been rewritten to improve the comprehension. However, we have to write at least 4000 words, following journal guidelines.

  1. Prisma diagram should be included

Author response: thanks for the suggestion, PRISMA diagram has been added.

  1. It is not clear which disoders do the author classify as inflammatory, if morphea is inflammatory other autoimmune disorders should be as well

Author response: thanks for the suggestion, Morphea is both an autoimmune and inflammatory disorder. For this reason, we do not particularly focus the attention on this disease.

  1. The work is incomplete, as papers relevant to the topic have not been cited, such as :

Majid I, Mearaj S. Sweet syndrome after Oxford-AstraZeneca COVID-19 vaccine (AZD1222) in an elderly female. Dermatol Ther. 2021 Nov;34(6):e15146. doi: 10.1111/dth.15146. Epub 2021 Oct 7. PMID: 34590397; PMCID: PMC8646808.

Agaronov A, Makdesi C, Hall CS. Acute generalized exanthematous pustulosis induced by Moderna COVID-19 messenger RNA vaccine. JAAD Case Rep. 2021 Oct;16:96-97. doi: 10.1016/j.jdcr.2021.08.013. Epub 2021 Aug 27. PMID: 34466640; PMCID: PMC8393513.

Paolino G, Di Nicola MR, Cantisani C, Mercuri SR. Pityriasis rosea infection in a COVID-19 patient successfully treated with systemic steroid and antihistamine via telemedicine: Literature update of a possible prodromal symptom of an underlying SARS-CoV-2 infection. Dermatol Ther. 2021 Jul;34(4):e14972. doi: 10.1111/dth.14972. Epub 2021 May 24. PMID: 33993616; PMCID: PMC8209955.

Cantisani C et al. Cutaneous Reactions to COVID-19 Vaccines in a Monocentric Study: A Case Series. J Clin Med. 2022 Jun 30;11(13):3811. doi: 10.3390/jcm11133811. PMID: 35807096; PMCID: PMC9267144.

Camargo Coronel et al. Dermatomyositis post vaccine against SARS-COV2. BMC Rheumatol. 2022 Apr 1;6(1):20. doi: 10.1186/s41927-022-00250-6. PMID: 35361289; PMCID: PMC8970647.

Author response: thanks for the suggestion, all these references have been added.

  1. The selection of the searched keywords seem a bit random

Author response: thanks for the suggestion, searched keywords have been rewritten.

  1. In is not questioned if a vaccine reaction is reported somewhere it could be not related to the vaccine, as the majority of the population received vaccines, causality is sometimes hard to prove

Author response: thanks for the observation, This topic has been reported in the limitation section.

  1. There should be more comprehensive tables and figures, as overall it is difficult to draw the main findings of the paper

 Author response: thanks for the observation. However, we preferred reporting specific tables for each disease if there was a consistent number of cases in order to avoid confusing table where several data are reported. Moreover, there are a lot of manuscript reporting comprehensive tables and we preferred to made something different.

Minor issues:

  1. Keywords do not include Covid-19
  2. Numbers should be sound, such as in the abstract, why skin reactions of about 550 patients were included, is not there an exact number?

 Author response: thanks for the observation. Keywords have been corrected. However, the exact number of cases cannot be reported as some patients were may be counted twice and other misses, as well as some manuscripts do not report the exact number of cases.

Reviewer 4 Report

The manuscript represents an interesting approach of analyzing dermatologic disorders post-vaccination. Even though the spectrum of different diseases is interesting and has clinical significance, the number of cases and the vaccines that actually may induce or aggravate the lesions is unclear. It is probably more important to perform a meta-analysis with a more interesting approach. It is also important to address the issue of the time of the clinical manifestation since one may infer that the immune response produced by the vaccines is temporary. Consequently, the decrease in clinical manifestation may occur after a period close to 6 weeks. It will also be interesting to discuss the cases of patients with dermatological manifestations, how the disease has progressed and what could be the more appropriate medical approach. Finally, according to the general statistics published, dermatologic manifestations after the first vaccination are close to one case per million; however, it is unclear if repetitive dosis of vaccines may decrease these event. This issue should be addressed by the authors.

The English language, in general, is readable. There are minor grammatical mistakes and typo errors

Author Response

Dear Reviewer, 

Thank you for your review. 
The review was by invitation so meta-analysis is not possible, however, we added a prism figure to make the whole text more understandable. We corrected spelling errors as you requested and discussed the cases of vaccination after the first dose and after the second dose. 
Regarding treatment we do not think it is important as there are different dermatologic manifestations treated and the treatment may depend on various factors as well as the patient's comorbidities, plus many reports are single case reports or small case series so there was not really reported treatment. The lack of guidelines is definitely an important point and we have included it in the limitations paragraph. 

Best regards

Round 2

Reviewer 3 Report

The manuscript has improved considerably following this major revision.

However, there are some further minor things to be clarified by the authors:

1. The term of inflammatory disease should be clarified, as it is a bit strange that infectious conditions such as herpes zoster are included

2. In the PRISMA flowchart, it is missing why they have excluded certain studies

3. I find it a bit too strong statement that urticaria is the second most common reaction after vaccinations, as while it could be reported the second most commonly, it is hard to know the exact prevalence of all reactions

Author Response

The manuscript has improved considerably following this major revision.

Author response: thanks for appreciating our manuscript

However, there are some further minor things to be clarified by the authors:

1. The term of inflammatory disease should be clarified, as it is a bit strange that infectious conditions such as herpes zoster are included

Author response: thanks for the comment. In order to improve the manuscript, the title has been changed.

2. In the PRISMA flowchart, it is missing why they have excluded certain studies

Author response: thanks for the comment, PRISMA flowchart has been improved as suggested.

3. I find it a bit too strong statement that urticaria is the second most common reaction after vaccinations, as while it could be reported the second most commonly, it is hard to know the exact prevalence of all reactions

Author response: thanks for the suggestion. Urticaria section has been improved as suggested.

Reviewer 4 Report

The authors made only partial changes to the manuscript. Figure 1 is important as well as minor details in the language. In my opinion, there are assumptions which, without the support of meta-analysis, have to be taken just as suggestive and not as definite proposals, specially in the discussion

The English language is understandable. Minor typo mistakes were observed.

Author Response

The authors made only partial changes to the manuscript. Figure 1 is important as well as minor details in the language. In my opinion, there are assumptions which, without the support of meta-analysis, have to be taken just as suggestive and not as definite proposals, specially in the discussion

Author response: thanks for the comment. Figure 1 has been improved as suggested. In our study, we performed a review of the current literature investigating cutaneous reactions following Covid-19 vaccination. Agreeing with you, without the support of meta-analysis, have to be taken just as suggestive and not as definite proposals. Therefore, we state this limitation of the study.